# Inferior and Coordinate Distillation for Object Detectors

**DOI:** 10.3390/s22155719

**Published:** 2022-07-30

**Authors:** Yao Zhang, Yang Li, Zhisong Pan

**Affiliations:** School of Command and Control Engineering, Army Engineering University of PLA, Nanjing 210007, China; zhangyao111019@163.com (Y.Z.); solarleeon@outlook.com (Y.L.)

**Keywords:** inferior distillation, knowledge distillation, object detection, refine module

## Abstract

Current distillation methods only distill between corresponding layers, and do not consider the knowledge contained in preceding layers. To solve this problem, we analyzed the guiding effect of the inferior features of a teacher model on the coordinate feature of a student model, and proposed inferior and coordinate distillation for object detectors. The proposed method utilizes the rich information contained in different layers of the teacher model; such that the student model can review the old information and learn the new information, in addition to the dark knowledge in the teacher model. Moreover, the refine module is used to align the features of different layers, distinguish the spatial and channel to extract attention, strengthen the correlation between the features of different stages, and prevent the disorder caused by merging. Exclusive experiments were conducted on different object detectors. The results for the mean average precision (mAP) obtained using Faster R-CNN, RetinaNet, and fully convolutional one-stage object detector (FCOS) with ResNet-50 as its backbone were 40.5%, 39.8%, and 42.8% with regard to the COCO dataset, respectively; which are 2.1%, 2.4%, and 4.3% higher than the benchmark, respectively.

## 1. Introduction

With the rapid development of deep learning, convolutional neural networks (CNNs) are being increasingly implemented in various applications such as autonomous driving and pedestrian recognition, among others. However, the computational costs and delays are increasing, thus limiting the applicability of CNNs [1]. To make CNNs more suitable for layouts on mobile devices and ordinary vision sensors, a more compact and real-time CNN is needed [2].

To overcome this limitation, knowledge distillation was first proposed by Hinton [3], in that the dark knowledge in the large model (i.e., teacher model) should be used to better train the compact model (i.e., student model), such that the compact model can obtain an equivalent generalization capacity to the large model. Hinton considered the output of the teacher model after using softmax as a soft label, and guided the training of the student model using the soft label as well as ground truth, which can be expressed as follows:
(1)$$L=\alpha L_{origin}+\beta L_{soft}$$
where $L_{origin}$ represents original training loss of student model, and $L_{soft}$ represents the distillation loss using the soft label.

Different from the logits distillation proposed by Hinton, Fitnets [4] proposed feature distillation, which involves the extraction of the middle-layer features of the teacher model to guide the training of the student model, thus causing the student model to imitate the features of the teacher model.

Knowledge distillation has been successfully applied to image classification [5,6,7,8]. However, its application to object detection, which is a major computer vision task, requires further investigation.

Compared with image classification, object detection models are required to demonstrate simultaneous classification and location capacities. Thus, they are more complex and contain more information. It is therefore relatively more difficult to transfer knowledge from the teacher detector to the student detector.

Chen et al. [9] first applied knowledge distillation to object detection by combining logits distillation and feature distillation. However, significant noise was introduced due to the significant imbalance between positive and negative samples in object detection. To solve the abovementioned problem, fine-grained feature imitation (FGFI) [10] and task adaptive distillation framework (TADF) [11] utilize a fine-grained mask and Gaussian mask, respectively, to select the distillation area, such that the distillation area is concentrated around the target. Guo et al. [12] demonstrated that only using the background area for distillation can achieve the same performance as that using the foreground area, which indicates that foreground area distillation helps to improve the accuracy of the results; whereas, background area distillation helps to reduce the probability of false detection.

However, previous research was focused on the problem of locating the key area of distillation, and did not consider the selection of the objects to be distilled. The distillation object is the key point in distillation. In previous research, it was hypothesized that distillation is only carried out between the corresponding layers as a default setting. In addition, layers lower-level than the current layer were not considered, although they may help to improve the distillation performance, i.e., causing the third-stage output of the student model to mimic the first- and second-stage outputs of the teacher model.

This paper first summarizes the distillation methods developed in previous research. As shown in Figure 1, in previous research, distillation was considered only between corresponding layers; thus indicating that only the same level of knowledge was utilized. This type of distillation is referred to as coordinate distillation in this manuscript. However, the layers before the coordinate layer, which are referred to as inferior layers in this manuscript, can facilitate the training of the student model.

For verification, an experiment was conducted on RetinaNet. The backbone of teacher model was ResNet101, and backbone of student model was ResNet50. The visualization results are shown in Figure 2.

In an attention heatmap, darker areas receive greater attention by the model. As can be seen from Figure 2, the attention heatmap on the right is more concentrated on the target (the woman playing tennis), which indicates that the student model simultaneously guided by the inferior layer and coordinate layer directs increased attention to the target area; thus, demonstrating an improved model performance. In particular, knowledge in inferior layers can facilitate the training of the student model during distillation as the corresponding layer; however, superior layers can negatively impact the learning curve. As expected, when students review previous information, they can learn the new information by re-learning the old.

Based on the above, this paper proposes inferior and coordinate distillation (ICD) for object detectors, which is a combination of inferior distillation and coordinate distillation. In addition, this paper proposes a refine module to merge features from different stages and strengthen their semantic expression, considering that the merging of features from different stages may distort the semantic expression. This module aligns inferior and coordinate features to the same shape and merges them. Thereafter, it adopts a refine module, extracts the correlation matrix of spatial and channel separately, and superimposes them onto the original feature map to obtain the attention maps. By fusing the two attention maps via a convolution layer, a refined feature with the same size as the original feature image can be obtained.

Extensive experiments were conducted, as presented in Section 4, which demonstrate the effectiveness of the proposed method. In a nutshell, the contributions of this paper are as follows:
We demonstrate that inferior distillation facilitates the learning of the student model. If the inferior layer is used to guide the learning, it will result in certain improvement.We propose inferior and coordinate distillation for object detection models, which enables the student to simultaneously focus on the inferior information and the information contained in the coordinate layer of the teacher model. This indicates that multi-level information is utilized to guide the learning of the student model.Experiments conducted on various detectors, including one-stage, two-stage, and anchor-free detectors verified the effectiveness of the proposed method.


## 2. Related Works

### 2.1. Object Detection

Object detection is a basic computer vision task. The main objective is to figure out the category and location information of one or more objects in an input image. Object detection models based on deep learning can be classified as one-stage detector [13,14,15] and two-stage detectors [16] based on the detection process.

The object detection model generally consists of three parts: a backbone network to extract semantic features; a neck network to fuse multi-scale information, and a detection head to output classification and location information. In addition, compared with the one-stage detector, the two-stage detector contains a region proposal network (RPN) network for the generation of proposals. Although the RPN network can obtain superior results, it exhibits an increased computing overhead and delay.

One-stage detectors can be divided into two categories based on whether the anchor is pre-set. The anchor box in a one-stage detector replaces the RPN network in two-stage detectors. However, the anchor boxes are domain-specific and less generalized. Anchor-free detectors can directly predict the category and location of a target, and demonstrate a lower computational overhead and delay.

Although the model structures of these three types of detectors are different, the proposed knowledge distillation algorithm can be applied to all the above-mentioned detectors based on feature extraction from the neck network.

### 2.2. Knowledge Distillation

Knowledge distillation (KD) refers to a method that transfers knowledge from a large model (i.e., teacher model) to a small model (i.e., student model), and can achieve a high performance without any additional cost in forward inference. The advantage of knowledge distillation is that the structure of the student model is unchanged after distillation. In addition, knowledge distillation and pruning are orthogonal and can be combined to compress the model.

Knowledge distillation was first proposed by Hinton [3], which utilizes the soft label of the teacher model to better train the student model. Fitnets [4] confirmed that the feature of intermediate layers can guide the training of the student network and proposed hint learning. Moreover, KD is commonly used in image classification, and requires further investigation with respect to object detection.

Chen et al. [9] first applied knowledge distillation to object detection by combining logits distillation and feature distillation; however, considerable noise led to a decrease in performance. Thereafter, significant research was conducted on KD for object detectors; with focus on the selection of the distillation area [12,17], or on prediction-guided imitation [18,19] to avoid hand-craft region selection.

In summary, current methods conduct distillation between the corresponding layers and consider it as a default set. However, the proposed method for object detectors includes the guide of inferior layers during distillation.

## 3. Materials and Methods

To make the best of the knowledge from different stages, the proposed method causes the student coordinate layer to simultaneously mimic the inferior and coordinate layers of the teacher model. In summary, this paper proposes a simple scheme to implement a distillation method utilizing inferior and coordinate information. The overall framework of the proposed method is shown in Figure 3. We utilized a refine module to prevent disorder during feature merging. In the refine module, we first align and merge the features from different stages, and then strengthen the semantic expression of the merged feature.

For applicability to various types of object detection models, the proposed method mainly uses the multi-scale feature map contained in the neck network. These feature maps contain different levels of semantic information, and the extracted dark knowledge can effectively improve the student model performance.

### 3.1. Refine Module

Although inferior and coordinate distillation improve the effect of distillation, a significant computational cost is incurred. To simplify the calculation, a refine module is proposed, which merges the coordinate feature and the inferior features, such that the student model can progressively learn the knowledge contained in the teacher model.

Different from the convolutional block attention module (CBAM) [20] and focal and global distillation (FGD) [17], the proposed refine module utilizes the residual structure to merge features from different stages, thus reducing the calculation complexity. Moreover, it separately extracts spatial and channel attention, and fuses them via a convolution layer. The structure of the refine module is shown in Figure 4. Where A, B, C and D denotes feature maps, S and X denotes the spatial and channel relationship intensity matrixs, E and F denotes the spatial and channel attention maps.

The refine module first adjusts the output of the preceding refine module to the same shape as that of the coordinate feature map, and then merges them via a convolution layer. Thus, the output size of refine module is the same as that of the coordinate feature map. Thereafter, channel and spatial attention maps are extracted, respectively. Finally, the two attention maps are combined to obtain a refined feature map.

When extracting spatial attention map, as long as the spatial information in the attention map is enhanced, the spatial correlation between the features of different stages can be strengthened. The process is as follows. First, feed Feature A (A∈ℝC×H×W) into a convolution layer to generate Matrixes B, C, and D, respectively, and reshape them to the shape of C × N; where N = H × W represents the number of pixels. Thereafter, a matrix multiplication is performed between C and the transpose of B, and a softmax layer is applied to the output of the matrix multiplication. Thus, the spatial relationship intensity matrix S indicating the relationship between any two points in Feature A is obtained as follows:(2)Sij=exp(BiCj)∑i=1Nexp(BiCj)
where N = H × W. Moreover, H and W are the height and width of the feature map, respectively; and Sij represents the spatial relationship between the *i*th pixel and the *j*th pixel. With an increase in Sij, the correlation between the two pixels increases.

A matrix multiplication is then applied between Matrix D and the transpose of Matrix S to selectively enhance or suppress the expression of features and reshape the result to the original shape. After applying an element-wise sum to the output of multiplication and the original feature A, we obtain a spatial attention map E as follows:(3)$$E_{j}=\alpha \sum_{i=1}^{N}(S_{ij}D_{i})+A_{j}$$
where *α* is a learnable scale coefficient set initially to 0. It can be seen that the spatial attention map *E* is the weighted sum of the corresponding positions in the feature map A, which establishes the dependency of global features.

In summary, this step selectively aggregates the spatial features of each location by a weighted sum of the features at all locations.

The procedure for extracting channel attention is similar to that for spatial attention. First, the channel relationship intensity matrix *X* is calculated as follows:(4)$$X_{ij}=\frac{\exp(A_{i}A_{j})}{\sum_{i=1}^{C}\exp(A_{i}A_{j})}$$
where *C* is the number of channels, and $X_{ij}$ indicates the influence of the *i*th channel to the *j*th channel.

A matrix multiplication is then performed between Feature A and the transpose of Matrix *X*, and the result is reshaped to the original shape. Thereafter, an element-wise sum is applied between matrix *X* and original feature A. Thus, the channel attention map F is obtained:(5)$$F_{j}=\beta \sum_{i=1}^{C}\left(x_{ji}A_{i}\right)+A_{j}$$
where *β* is a learnable scale coefficient similar to *α*. It can be seen that the channel attention map F is the weighted sum of each channel feature and the original feature A.

Finally, the spatial attention map E and the channel attention map F are fused via an element-wise sum and a convolution layer. Therefore, the relationship of the spatial and channel dimensions in Feature map A can be enhanced based on the merging of features from different stages, and the distillation effect is improved.

After the refine module, student features can be represented as follows:(6)$$\begin{array}{c} F_{i}^{S^{\prime }}=g(F_{i}^{S},F_{i+1}^{S^{\prime }})=g(F_{i}^{S},F_{i+1}^{S},F_{i+2}^{S^{\prime }})\\ \;=g(F_{i}^{S},F_{i+1}^{S},\cdots ,F_{M}^{S}) \end{array}$$
where *g*(·) represents the operation of the refine module.

### 3.2. Overall Loss

In general, the vanilla feature distillation of a mono-layer can be formalized as follows:(7)$$L_{fea}(F^{T},F^{S})=\frac{1}{CHW}{\sum_{k=1}^{C}\sum_{i=1}^{H}\sum_{j=1}^{W}\left(F_{k,i,j}^{T}-f(F_{k,i,j}^{S})\right)}^{2}$$
where *C*, *H*, and *W* represent the number of channels, height, and width of the feature map, respectively; $F^{T}$ and $F^{T}$ represent the feature maps of the teacher and student models, respectively; and f(⋅) represents an adaptive layer that aligns student and teacher features.

Multi-layer feature distillation involves the selection of multiple pairs of corresponding layers in the teacher and student models, in addition to the calculation of the separate distillation losses and their summation:(8)$$L_{dis\_vanilla}=\sum_{m=1}^{M}L_{fea}(F_{m}^{T},F_{m}^{S})$$
where *M* is the number of selected layers.

The vanilla knowledge distillation is generally only carried out between the corresponding layers, and the distillation effect is not considered due to the knowledge transfer between different layers.

In contrast, the proposed method utilizes coordinate information and inferior information. The student features are simultaneously obtained from inferior features and corresponding features of the teacher model, as shown in Figure 1b. The loss associated with inferior and coordinate distillation can be calculated as follows:(9)$$L_{dis}=\sum_{m=1}^{M}\sum_{n=1}^{m}L_{fea}(F_{n}^{T},F_{m}^{S})$$

Thus, the high-level student features can “observe” all the preceding teacher features, and the student model can progressively learn the knowledge from the teacher model. With the addition of the refine module to simplify the calculation, the loss can be formalized as follows:(10)$$\begin{array}{c} L_{dis}=\sum_{m=1}^{M}\sum_{n=1}^{m}L_{fea}(F_{m}^{T},F_{m}^{S})\\ \;=\sum_{m=1}^{M}L_{fea}(F_{m}^{T},F_{m}^{S^{\prime }})\\ \;=\sum_{m=1}^{M}L_{fea}(F_{n}^{T},g(F_{m}^{S},F_{m+1}^{S},\cdots ,F_{M}^{S})) \end{array}$$

In summary, the total training loss of the student model is expressed as follows:(11)$$L=L_{origin}+L_{dis}$$
where $L_{origin}$ refers to the original training loss of the object detection model. Since the distillation loss only utilizes the feature maps in the neck network, the proposed method is suitable for one-stage and two-stage object detection models.

## 4. Results and Discussion

### 4.1. Dataset and Measurement

We evaluated the proposed knowledge distillation method using the COCO dataset [21], which contains more than 220 k labeled images. The COCO dataset is a large-scale dataset with 80 object categories such as pedestrians and vehicles, with a total of 1.5 million targets. In all experiments, training dataset and test dataset are split in 8:2, which means that 170 k images are used for training and 5 k images are used for test.

In the experiment, the performance of distillation was evaluated with respect to the mean Average Precision (mAP) of the detectors. AP indicates the average precision under different IoU of a given category, and mAP is the mean value of AP of different categories. All mAP reported are obtained on the test dataset.

### 4.2. Details

All experiments were performed using two GTX 2080Ti GPUs. The experiment was based on MMdetection [22], which is an open-source object detection toolbox based on PyTorch. To confirm the generality of the proposed method, three mainstream object detection models were selected for experiments, namely, RetinaNet [13], which is an anchor-based one-stage detector; FCOS [14], which is an anchor-free one-stage object detector; and Faster R-CNN [16], which is a two-stage object detector.

We trained all the detectors for 24 epochs using SGD optimizer, wherein the momentum was 0.9 and the weight decay was 0.0001.

### 4.3. Main Results

We conducted experiments on Faster R-CNN, RetinaNet, and FCOS; and compared the results with four other current mainstream knowledge distillation methods for object detectors, i.e., FGFI [10], general instance distillation (GID) [23], rank mimicking and prediction-guided feature imitation (RMPFI) [18], and FGD [17]. Given that FGFI is not suitable for the anchor-free detectors, we only used the Faster R-CNN and RetinaNet for comparison.

In the experiment, the backbone of the student model was ResNet-50, and the backbone of the teacher model was ResNet-101. The experimental results are shown in Table 1, where T and S denote the teacher and student models, respectively.

It can be seen from Table 1 that the proposed method demonstrated a high performance on different object detectors, and the accuracy was improved to an extent. For example, the mAP of Faster R-CNN, RetinaNet and FCOS were improved by 2.1%, 2.4%, and 2.3%, respectively, after distillation, which is superior to several SOAT distillation methods.

To illustrate the influence of using different teacher models to distill the same student model, we experimented on RetinaNet and FCOS with ResNext-101 and ResNet-101 as the backbone of the teacher model, and ResNet-50 as the backbone of student model. The results were as follows.

It can be seen from Table 2 that when the backbone of the teacher model changed from ResNet101 to ResNext101, the mAP of the FCOS after distillation decreased. This indicates that an improved teacher model does not always lead to an improved student model. This may be because the student model exhibits a sufficient capacity for feature extraction. Adding the supervision of the teacher model leads to the over-fitting of the student model.

### 4.4. Ablation Study

Ablation experiments were conducted, in which the ablation components were added individually to measure their effects. We used FCOS-ResNet101 as the teacher model, and FCOS-ResNet50 as the student model. The results are shown in Table 3, where the symbol ‘√’ means this module has been added into distillation.

It can be seen from Table 3 that after adding inferior information into distillation, the distillation performance increased by 1.6%, which is equivalent to the increase by coordinate distillation (i.e., the vanilla distillation). The refine module contributed a gain of 0.9%, thus verifying its capacity to enhance the semantic expression in the spatial and channel dimensions.

In summary, compared with the canonical distillation, the proposed method can transfer more dark knowledge from the teacher model to the student model.

## 5. Conclusions

This paper highlights that inferior information in the teacher model can guide the learning of the student model. Hence, this paper proposes a distillation method for object detection. First, the inferior and coordinate features of the teacher model are utilized to guide the learning process of the student model. Thereafter, the merged features are strengthened via the refine module. In the refine module, spatial and channel attention maps are separately extracted and fused to eliminate the negative influence of merging features from different stages.

The comparative experiments revealed that the proposed ICD method is simple, effective, and can be applied to various object detectors, including one-stage detectors with and without anchor boxes, as well as two-stage detectors.

The proposed method demonstrates that inferior and coordinate distillation for object detectors can effectively transfer the dark knowledge contained in the teacher model to the student model and improve the performance of the student model. Utilizing inferior information is a future research focus of knowledge distillation. In particular, we will investigate methods for the utilization of the knowledge contained in the detection head, and comprehensively analyze the influence of the detection head on distillation.

Deep learning is widely used in realistic applications. However, its implementation consumes large amounts of energy. The research of this paper can obtain a more efficient student model with less energy consumption, which promotes the development of ‘green’ deep learning.

## Figures and Tables

**Figure 1 sensors-22-05719-f001:**
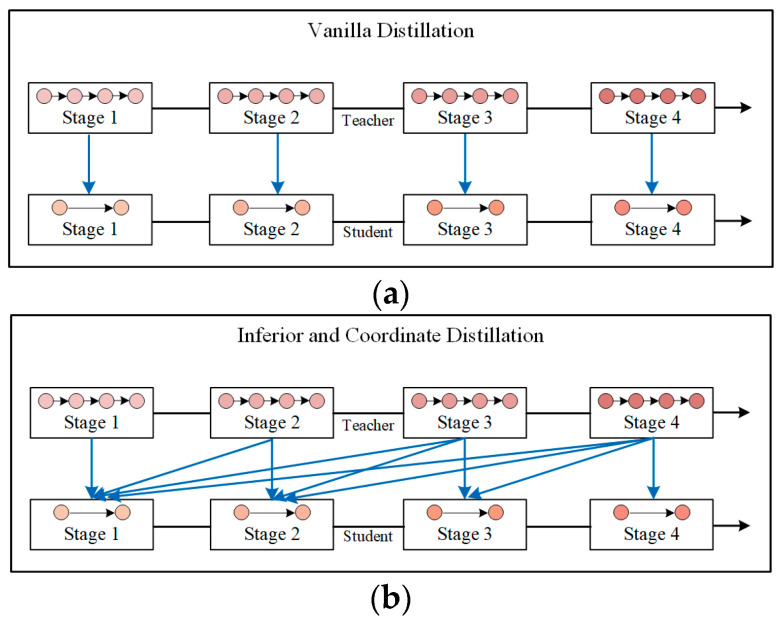
Comparison between vanilla distillation (**a**) and inferior and coordinate distillation (**b**).

**Figure 2 sensors-22-05719-f002:**
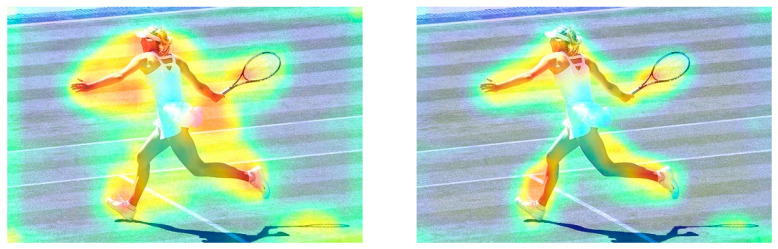
Visualization of the attention heatmap obtained from different distillation methods: utilizing (**left**) only the coordinate knowledge during distillation and (**right**) both coordinate knowledge and inferior knowledge.

**Figure 3 sensors-22-05719-f003:**
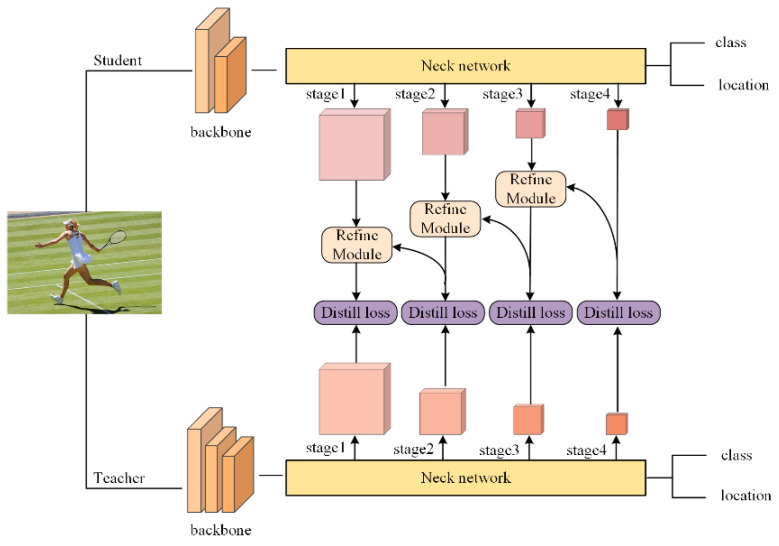
An illustration of the ICD method.

**Figure 4 sensors-22-05719-f004:**
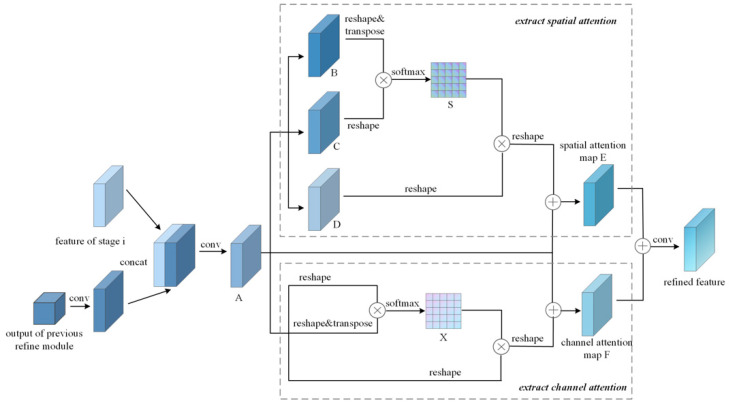
Structure of the refine module.

**Table 1 sensors-22-05719-t001:** Performance comparison between different distillation methods using the COCO2017 dataset. The bold means the best result.

Method	mAP/%
Faster R-CNN_ResNet50 (S)	38.4
Faster R-CNN_ResNet101 (T)	39.8
FGFI	39.3 (+0.9)
GID	40.2 (+1.8)
RMFPI	40.5 (+2.1)
FGD	40.4 (+2.0)
**Ours**	**40.5 (+2.1)**
RetinaNet_ResNet50 (S)	37.4
RetinaNet_ResNet101 (T)	38.9
FGFI	38.6 (+1.2)
GID	39.1 (+1.7)
RMFPI	39.6 (+2.2)
FGD	39.6 (+2.2)
**Ours**	**39.8 (+2.4)**
FCOS_ResNet50 (S)	38.5
FCOS_ResNet101 (T)	40.8
GID	42.0 (+4.5)
RMFPI	42.3 (+4.8)
FGD	42.7 (+4.2)
**Ours**	**42.8 (+** **4.3)**

**Table 2 sensors-22-05719-t002:** Effects of different teacher models on distillation.

	Detectors	FCOS	RetinaNet
Teacher			
ResNet-50		38.5	37.4
ResNet-101		42.8	39.8
ResNext-101		42.5	40.2

**Table 3 sensors-22-05719-t003:** Ablation study of ICD.

Method	Coordinate Distillation	Inferior Distillation	Refine Module	mAP/%
FCOS-ResNet101Distill FCOS-ResNet50				38.5
	√			40.1 (+1.6)
	√	√		41.7 (+3.2)
	√	√	√	**42.8** **(+** **4.3** **)**

## Data Availability

The study did not report any data.

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
