# Peer review of "Inferior and Coordinate Distillation for Object Detectors"

_sensors, 2022, doi:10.3390/s22155719_

Round 1

Reviewer 1 Report

The work proposed inferior and coordinate distillation for object detectors. The paper is interesting, well written and structured. As suggestions for improvement, I highlight the points below:

1)      The first paragraph of the introduction appears to be out of standard formatting. I suggest correcting this error.

2)      2) Some quotes are wrong, such as “Hinton [2].” I suggest a general review of the references.

3)      3) The Literature Review is too short and should be extended, presenting recent studies dealing with Intelligent Sensors, citing works such as:

https://link.springer.com/chapter/10.1007/978-3-030-76310-7_9

4)      The results and conclusions must be improved, explaining the main contributions of the paper to the scientific community and society.

Reviewer 2 Report

Knowledge distillation is a hot topic in large-scale machine learning and deep learning-based ones, which aims to capture and “distilling” the knowledge in a complex machine learning model or an ensemble of models to form a smaller single model. It is very interesting for the proposed method in using the information of the teacher model for object detectors. This writting is good, and the experimental demonstration is also sufficient. 

Author Response

Thank you very much!

Reviewer 3 Report

The paper proposed a distillation model for an object detection task that acquired knowledge from subsequent layers of a teacher model to train each preceding layer of the student model. The method is proven effective when compared to a traditional knowledge distillation method where only the output from a single layer is used to train the corresponding layer of a student model on several models. Overall, the proposed model is interesting and the experiments to validate the claim are performed thoroughly. 

Some minor comments are as follows.

- The description and formula expression for mAP (mean Average Precision) shall be given.

- In figure 3, one annotation is still in Chinese.

- In table 2, '\' shall be replaced by the name of the respective teacher model.

- How the training and test set are split and whether the mAP result is obtained on training or test set shall be indicated.

Round 2

Reviewer 1 Report

The authors complied with the suggestions for improvement. Therefore, I recommend accepting the paper.